# Robust all-optical single-shot readout of nitrogen-vacancy centers in diamond

Dominik M. Irber [1,2], Francesco Poggiali [1,2], Fei Kong [3], Michael Kieschnick[4], Tobias Lühmann[4], Damian Kwiatkowski[5], Jan Meijer[4], Jiangfeng Du [3], Fazhan Shi [3] & Friedemann Reinhard [1,2,6 ✉]

High-fidelity projective readout of a qubit's state in a single experimental repetition is a prerequisite for various quantum protocols of sensing and computing. Achieving single-shot readout is challenging for solid-state qubits. For Nitrogen-Vacancy (NV) centers in diamond, it has been realized using nuclear memories or resonant excitation at cryogenic temperature. All of these existing approaches have stringent experimental demands. In particular, they require a high efficiency of photon collection, such as immersion optics or all-diamond micro-optics. For some of the most relevant applications, such as shallow implanted NV centers in a cryogenic environment, these tools are unavailable. Here we demonstrate an all-optical spin readout scheme that achieves single-shot fidelity even if photon collection is poor (delivering less than $10^3$ clicks/second). The scheme is based on spin-dependent resonant excitation at cryogenic temperature combined with spin-to-charge conversion, mapping the fragile electron spin states to the stable charge states. We prove this technique to work on shallow implanted NV centers, as they are required for sensing and scalable NV-based quantum registers.

[1] TU München, Walter Schottky Institut and Physik-Department, Am Coulombwall 4, 85748 München, Germany. [2] Munich Center for Quantum Science and Technology (MCQST), Schellingstraße 4, 80799 München, Germany. [3] CAS Key Laboratory of Microscale Magnetic Resonance & Department of Modern Physics, University of Science and Technology of China, 230026 Hefei, China. [4] Applied Quantum Systems, Felix-Bloch Institute for Solid-State Physics, University Leipzig, Linnéstraße 5, 04103 Leipzig, Germany. [5] Institute of Physics, Polish Academy of Sciences, al. Lotników 32/46, 02-668, Warsaw, Poland. [6] Institut für Physik, Universität Rostock, Albert-Einstein-Str 23, 18059 Rostock, Germany. ✉email: friedemann.reinhard@uni-rostock.de

Much of the popularity of (negatively charged) Nitrogen-Vacancy (NV⁻) centers in diamond is owing to the fact that the readout of their electron spin is straightforward, since fluorescence intensity correlates with the spin state[1]. However, this simple readout approach is highly inefficient because the relative contrast between the spin states is short-lived (approx. 250 ns) and low (about 30%)[2]. This corresponds to a single-shot signal-to-noise ratio (SNR) of 0.05 (0.03) for a count-rate of 200 kcps (50 kcps). Thus, averaging over several hundred to several ten thousand of experimental repetitions is necessary to read out the spin state with an SNR of 1.

One option to increase the single-shot SNR is spin-to-charge conversion (SCC)[3,4]. This readout approach maps the fragile spin state to the more robust charge state of the NV center, which can be read out optically with close to 100% fidelity even at room temperature[5]. This mapping is typically achieved by first shelving the spin $m_s = |\pm 1\rangle$ population to the meta-stable singlet state of the NV⁻ center and subsequently ionizing the NV⁻ center out of the triplet[3] or the singlet[4] state during the lifetime of the latter. So far, SCC has reached readout fidelities of up to 67%, limited by non-deterministic shelving to and storing in the singlet state.

More sophisticated schemes for spin readout have achieved single-shot readout, i.e. a single-shot SNR >1. A first method exploits repetitive readout from a nearby nuclear ancilla qubit[6]. This method requires a strong and carefully aligned magnetic field, and efficient photon collection for readout to succeed within the lifetime of the nuclear qubit.

A second scheme consists in tuning a narrow-linewidth laser in resonance to a cycling transition in the low-temperature excitation spectrum of the NV⁻ center[7]. In this configuration, the NV⁻ is spin-selectively excited and thus producing fluorescence only if its spin state matches the used optical transition. This gives a high contrast signal for a finite time, limited by spin depolarization due to laser illumination. Therefore, the scheme requires all-diamond micro-optics for efficient photon acquisition. This is in particular not available for NV centers close to a planar sample surface.

Here, we present a single-shot readout scheme that eliminates the need for sophisticated optics. The key of our approach is SCC at cryogenic temperature, where resonant excitation enables both high spin-selectivity of SCC and efficient readout of the charge state by poor collection optics. In detail, our protocol employs resonant excitation[8] to only excite the NV⁻ if its spin state matches the used optical transition, typically a spin $|0\rangle$ transition. Simultaneous illumination with a high-power 642-nm laser ionizes the NV⁻ from the excited state, while not causing internal excitation dynamics (Fig. 1a). Doing so, we lift the fidelity of SCC well above a single-shot SNR of 1 for a natural NV center microns deep in the diamond ('deep NV'), and to the single-shot threshold for a shallow implanted NV center closer than 100 nm to the diamond surface. The technique also promises to be robust against strong misaligned background magnetic fields.

## Results

All measurements were performed in a home-built confocal microscope. The sample is in a Helium flow cryostat and can be illuminated through an air objective with numerical aperture of 0.95 simultaneously with three independently gateable lasers: a narrow-band red laser tuned to a strong cycling transition starting from spin state $|0\rangle$ ('resonant laser'); a strong red diode laser for photoionization ('ionization laser'); and a green diode laser for initialization of the charge and (in some experiments) the spin state (Fig. 1b). Besides, the NV⁻ center can be excited by two gateable microwave (MW) sources, tuned to both directly allowed spin transitions within the ground state. A static

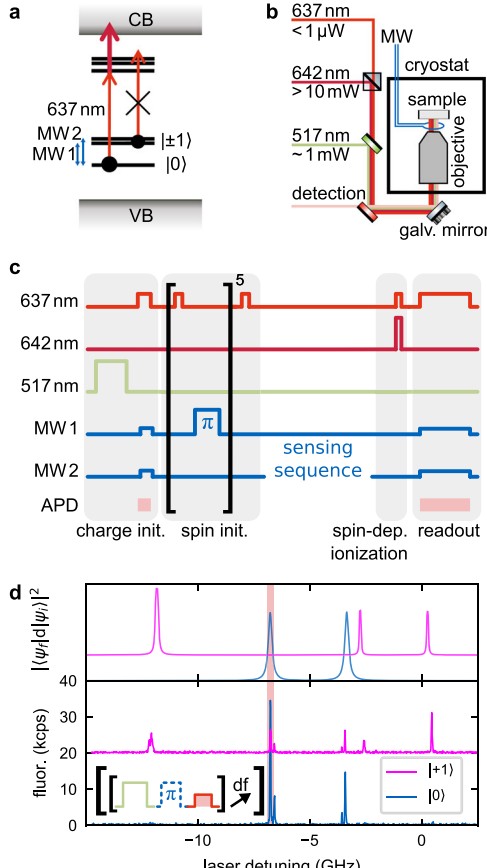

**Fig. 1 Main idea of the readout scheme. a** Energy levels (simplified) of an NV⁻ center in diamond. Initially, the NV⁻ center is in its (optical) ground state with spin state either $|0\rangle$ or $|\pm 1\rangle$. If the spin state is $|0\rangle$, a gated laser tuned to a $|0\rangle$ transition (637 nm; thin light red arrow) can excite the NV⁻ center, while spin $|\pm 1\rangle$ is protected against excitation. A second gated high-power laser (642 nm; bold dark red arrow) can ionize from the excited state. The blue arrows indicate the microwave (MW) transitions used below. CB/VB denote the conduction/valence band of the diamond host material. **b** Schematic of the setup. Three individually gated lasers can illuminate the sample, which is mounted in a flow cryostat. In addition, the sample can be driven by two MW frequencies. **c** Final pulse sequence. Red, dark red, and green correspond to gated 637 nm resonant, 642 nm, and 517 nm laser excitation. Blue refers to MW drive. During photon acquisition ('APD'; rose shade) for postselection and final data acquisition, cw MW excitation at both ground-state transitions is added to constantly mix the spin state during charge-state readout. **d** Lower panel: photoluminescence excitation (PLE) spectrum of the deep NV⁻ center that is also used for Figs. 2 and 3. Detuning is denoted from 637.20 nm. Off-axial strain is estimated to be 1.7 GHz. The inset shows the used pulsed sequence; 'df' indicates a change of the laser detuning. Upper panel: simulated spectrum according to Doherty et al.[10]. Red highlighting indicates the transition used below.

magnetic field of ~1 mT was applied, which was not aligned along the NV axis (Supplementary Section S.1).

These tools implement the final protocol (Fig. 1c). Its most crucial components are spin readout by (1) a highly spin-selective photoionization step ('spin-dep. ionization') implemented jointly by the resonant and the ionization laser and (2) low-power detection of the charge state by the resonant laser ('readout'), which is made agnostic to the spin state by a strong simultaneous MW drive ($T_{Rabi} = $~1 μs). Initialization of the charge state ('charge init.') is performed by the green laser and confirmed by a

spin-agnostic probe for later postselection. The spin state is initialized in $|+1\rangle$ by repeated resonant depletion of state $|0\rangle$ followed by emptying of the $|-1\rangle$ state by a MW pulse.

We first demonstrate the protocol on a deep natural NV center. At cryogenic temperatures, line narrowing allows different spin states to be individually addressed[9] and the NV$^-$ excited state reveals six sublevels. The measured spectrum (Fig. 1d) is well described by the model of Doherty et al.[10] (Fig. 1d upper panel) with a non-axial strain of 1.73 GHz. See Supplementary Section S.2 for more details. Two of them have $S_z$ character[8], thus having allowed cycling transitions from ground state spin $|0\rangle$, one of which ($-7$ GHz) serves as working transition for the red resonant laser. Note that $|0\rangle$ transitions are observed in the $|+1\rangle$ trace because of imperfect spin initialization in combination with the added MW $\pi$-pulse.

**Spin-state stability under optical pumping and spin-state initialization.** Driving the spin in state $|0\rangle$ on this selected transition with 56 nW ($0.08\ P_{sat}$; see Supplementary Fig. S.3 for saturation curves) induced fluorescence, which decayed to almost zero within 20 μs (Fig. 2a, upper panel, dark blue curve), as the spin is pumped from spin $|0\rangle$ to $|\pm1\rangle$ due to spin mixing[7]. During these 20 μs, we collected 0.17 photons on average. This low number compared to Robledo et al.[7] is due to the fact that we do not use any photonic structures, and precludes direct single-shot readout of the spin by resonant excitation[7] (fidelity of 52.8%; see Supplementary Fig. S.5 and Discussion S.4). This highly spin-selective fluorescence still enables benchmarking of the spin initialization. Using off-resonant excitation by a green laser for simultaneous charge and spin initialization, we obtain a mixture of $|0\rangle:|+1\rangle:|-1\rangle = (70 \pm 1):(13 \pm 1):(16 \pm 1)$ percent, consistent

with previous reports[2,11]. The most effective way to improve spin initialization is to pump on an optical spin $|\pm1\rangle$ transition. We decided for an experimentally simpler method; repeating optical depletion of the $|0\rangle$ transition and a $\pi$-pulse on the $|-1\rangle$ MW transition, which prepares the spin state $|+1\rangle$ with improved purity $(|0\rangle:|+1\rangle:|-1\rangle = (0 \pm 1):(88 \pm 2):(12 \pm 2)$ percent; Fig. 2a lower panel). See Supplementary Section S.5 for more details on the spin initialization.

**Charge-state stability and readout.** We also use the resonant laser to read out the charge state[12,13], which is in contrast to most SCC publications so far, which used an orange laser for that purpose. We counteract spin depletion by simultaneously applying cw MW excitation at both ground-state MW transitions to constantly mix spin population and in turn re-establish some population in $|0\rangle$. The charge state is stable under this combined excitation. Pumping on the transition with 13 nW ($<0.02\ P_{sat}$) plus cw MW, the NV$^-$ gets ionized after a second timescale (Fig. 2b). This can be seen as a sudden decrease in count rate to almost zero, because NV$^0$ has a higher-energy separation between ground and excited state, and is in turn protected against excitation by 637 nm. With the final readout power (56 nW, $0.08\ P_{sat}$), the charge state is stable for more than 10 ms (Fig. 2c). The photon statistics during a 1 ms readout pulse is presented in Fig. 2d. It displays clearly separated distributions for NV$^-$ and NV$^0$ events. In the final readout, events with $\geq 3$ clicks were assigned to be NV$^-$. NV$^-$ events have been produced by initialization using the green laser at 1.4 mW for 2 μs, initializing into the negative NV$^-$ charge state in $(46 \pm 1)\%$ of repetitions. Postselecting ($\geq 6$ clicks within 500 μs; keeping 37% of repetitions) on the charge state after the green illumination, as shown in Fig. 1c, improves initialization to NV$^-$ to $(99.7 \pm 0.7)\%$. Postselection also removes repetitions with severe spectral diffusion. NV$^0$ events are produced by first initializing NV$^-$ as described, and appending a 20-μs-long ionization pulse of 637 nm plus 642 nm, with cw MW added after 5 μs. Importantly, the high stability of the charge state under resonant excitation enables charge readout with near-perfect $((98.1 \pm 0.5)\%)$ fidelity using inefficient collection optics.

**Spin-dependent ionization.** The heart of our readout protocol is the spin-dependent ionization, which is a two-photon process. The second photon is provided by the strong (17 mW) red laser, red-detuned (642 nm) against the NV$^-$ zero phonon line (ZPL). It ionizes from the $^3E$ excited state on a fast (1 μs) time scale, but its energy is by itself insufficient to drive excitation into the $^3E$ state. Besides, it causes negligible stimulated emission back into the $^3A_2$ state[14]. Near-infrared (NIR) lasers fulfill these criteria, too[13], however we observed much less efficient ionization at 980 nm (see Supplementary Fig. S.9 and Discussion). Ionization is made spin-selective by simultaneously applying the resonant laser, which provides the first photon for excitation into the $^3E$ state. As this laser only excites spin $|0\rangle$, the spin population in the excited state and, hence, ionization is highly deterministic for spectrally well-separated transitions. Figure 3a shows the averaged fluorescence for charge-state readouts after the NV$^-$ center has been prepared in spin $|0\rangle$ or spin $|+1\rangle$ and spin-selectively ionized. Fluorescence is higher in the latter case, because spin $|+1\rangle$ is protected against resonant excitation and in turn against ionization. We expect the decay for spin $|+1\rangle$ for long ionization times to stem from residual excitation by the 642 nm laser. We optimized the ionization time for highest fluorescence contrast between the two preparations (resulting in 2 μs; contrast 7.5 kcps vs. 1.9 kcps). The resonant power was kept at 56 nm, as optimized for the readout. Figure 3b is the statistics of photon counts for the

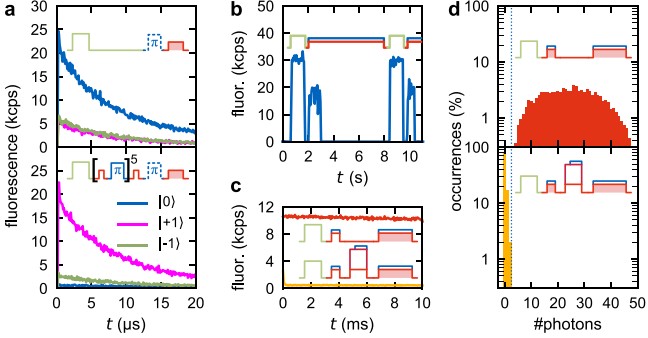

**Fig. 2 Spin- and charge-state stability of the deep NV center. a** Average fluorescence under optical pumping with a 637 nm laser being resonant to the main $|0\rangle$ transition. In the upper panel, the NV center was initialized to $|0\rangle$ with $(70 \pm 1)\%$ probability by a simple green laser pulse, followed by no MW excitation (blue) or by a $\pi$-pulse on either the $|0\rangle \leftrightarrow |+1\rangle$ or $|0\rangle \leftrightarrow |-1\rangle$ transition (magenta and green curve). The lower panel displays the same measurement after initializing to $|+1\rangle$ with $(88 \pm 2)\%$ probability using the explicit spin initialization protocol presented in Fig. 1c. The insets are sketches of the sequence used for the upper and lower panels. **b** Charge-state stability under excitation with the 637 nm laser plus cw MW excitation at both ground-state MW transitions. This was alternated with 1s of green repumping. These data are not averaged, but just one repetition. **c** Average fluorescence of the NV being preferentially in the charge states NV$^-$ (upper curve) or NV$^0$ (lower curve). These data (without using postselection) were taken simultaneously with the data presented in **d**. **d** Distribution of the number of fluorescence photons that were detected during 1 ms of readout. The upper panel displays the distribution directly after the charge-state initialization to NV$^-$, as shown in Fig. 1c, while the lower part includes a strong ionization pulse in between initialization and readout for conversion to NV$^0$. The dotted line indicates the threshold used for analysis. Note the logarithmic $y$ scale.

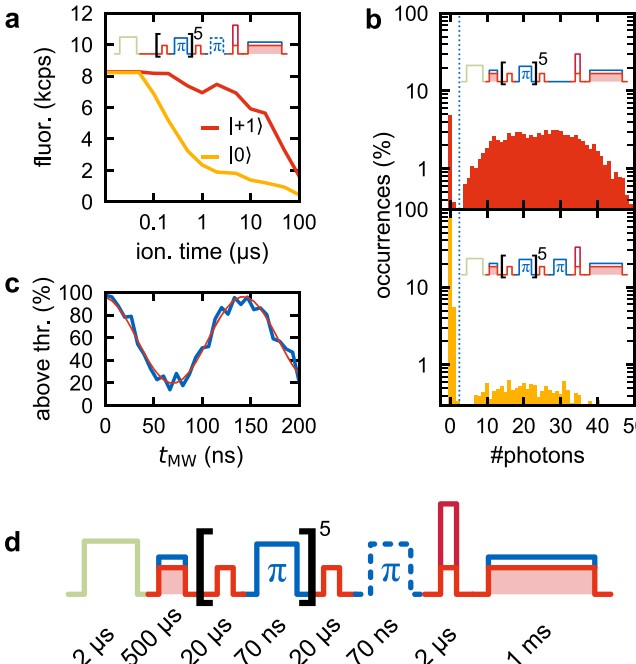

**Fig. 3 Spin-dependent ionization. a** Average fluorescence of the charge-state readout after charge initialization (without postselection), spin initialization according to Fig. 1c, and spin-dependent ionization with varying time. For **b** and **c**, 2 μs was used. **b** Distribution of the number of fluorescence photons that were detected during 1 ms of readout, after preparing the NV⁻ in either spin |+1⟩ or |0⟩ and spin-dependently ionizing it for 2 μs. The as-measured fidelity is (88.5 ± 0.5)%. The dotted line indicates the threshold used for analysis. Sketches of the used sequences are inset. **c** Rabi oscillation measured accordingly to panel **b**. The y-axis is the fraction of experimental repetitions with detected photon number during readout above threshold. The data were measured within about 11 s. The red line is a cosine fit. **d** Sequence as used for panel **b**. The color-code is explained in Fig. 1.

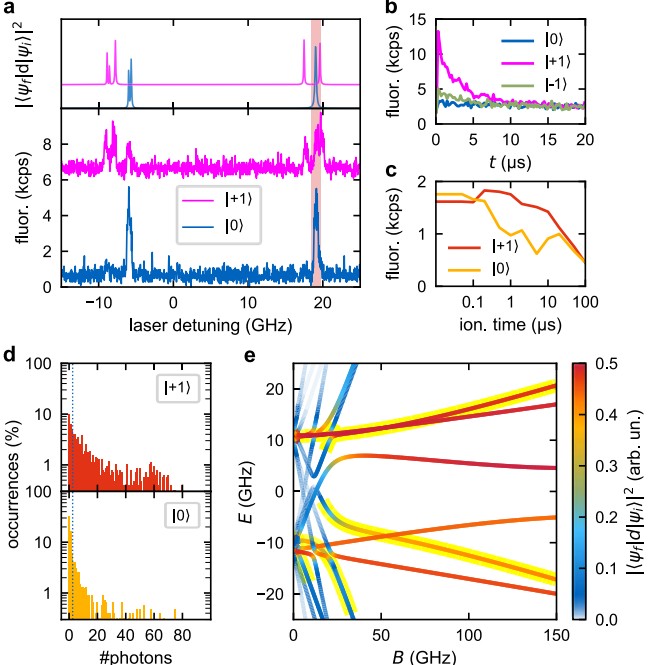

**Fig. 4 Data for a representative shallow implanted NV center (110 keV CN⁻). a** PLE spectra as measured (lower panel) and simulated (upper panel) similarly to Fig. 1d. Off-axial strain is estimated to be 12.6 GHz. The detuning is denoted from 637.20 nm. The 637 nm laser was tuned in resonance with the higher-energy transition at 19 GHz for all following measurements (red background). **b** Average fluorescence under optical pumping on the selected transition after explicit spin initialization according to Fig. 1c, as presented in Fig. 2a in the lower panel for the deep NV center. The NV⁻ center was initialized to |+1⟩ with (70 ± 1)% probability. The background after 20 μs stems from finite excitation of the close-by |±1⟩ transition. **c** Similar to Fig. 3a. Average fluorescence of the charge-state readout after charge initialization (without postselection), spin initialization in |+1⟩ or |0⟩ according to Fig. 1c, and spin-dependent ionization with varying ionization time. For **d**, an ionization time of 5 μs was used. **d** Similar to Fig. 3b. Distribution of the number of fluorescence photons for 5 ms of readout. The as-measured fidelity is (67.1 ± 0.9)%. **e** Simulation of the dipole transitions between optical ground and excited state for all three spin projection numbers for fixed strain-splitting of 20 GHz and varying magnetic field 20° off-axis. The transitions highlighted in yellow are the most pronounced spin |0⟩ transitions. The coloring is a measure for the transition strength. There are several parameter ranges with spectrally well-isolated transitions.

whole initialization–ionization–readout protocol (Fig. 3d) for both spin preparations. The measured end-to-end fidelity of the scheme is (88.5 ± 0.5)% (SNR 1.74 ± 0.06). Correcting for imperfect spin initialization (fidelity of (94.0 ± 0.9)%) and error of the MW π-pulse, the readout fidelity (comprising only the ionization and charge detection steps) is (96.4 ± 2.2)%, which corresponds to a single-shot SNR of 3.5 ± 1.2 (see Supplementary Section S.8). Reducing the readout time for the final charge state from 1 ms to 100 μs, the end-to-end fidelity is only slightly degraded from (88.5 ± 0.5)% to (82.3 ± 0.5)%. Importantly, the same performance could be achieved under strongly reduced photon flux (0.5 kcps instead of 50 kcps), if a 10-ms readout window is used (see Supplementary Section S.9). For a long (>10 ms) sensing sequence, this affords a speedup of 10³ (for 50 kcps) over standard readout and a factor of 20 over resonant excitation readout⁷ (see Supplementary Section S.4). For a short sequence in typical (50 kcps) conditions, our method is still as fast as the standard technique (Supplementary Section S.4). As an example, Fig. 3c shows Rabi nutations measured with 144 repetitions in 11 s, corresponding to a speed-up factor of one.

**Demonstration on a shallow implanted NV center.** Our method is applicable to 'shallow' NV centers less than 100 nm close to the diamond surface. Figure 4a–d presents data recorded on an ~70 nm deep center (110 keV CN⁻ implant; ref. [15]). Spectral lines are inhomogeneously broadened to (0.43 ± 0.02) GHz due to spectral diffusion (see Supplementary Section S.10). As the transitions are

also much weaker—yielding lower fluorescence—we compensate for this by increasing the resonant red laser power to 240 nW, which is just low enough to prevent pronounced effects by also exciting the close-by |1⟩ transition. Still, this close-by transition compromises spin initialization fidelity (Fig. 4b). Without post-selecting on the charge state, the useable contrast was best for 5 μs ionization time and yielded 0.6 kcps and 1.5 kcps for spin |0⟩ and |+1⟩, respectively (Fig. 4c). Including postselection on the charge state, the end-to-end single-shot fidelity, as measured in Fig. 4d, is (67.1 ± 0.9)%. Correcting for the non-perfect spin initialization and π-pulse results in a fidelity of (78.6 ± 2.5)%, corresponding to a single-shot SNR of 0.99 ± 0.13.

We note that sub-GHz optical linewidths have been reported for comparable implanted NV centers as shallow as 10 nm[16,17]. The protocol also promises to be resilient to high-strain environments, since it does not make use of the spin-selective intersystem crossing into the ¹A singlet state. It only requires a well-separated spin-selective transition. Simulations of the optical

**Table 1 Overview of the initialization and readout metrics for both NV centers presented in this letter.**

| | Deep natural NV center | Shallow implanted NV center |
|---|---|---|
| NV$^-$ fraction (%) | 99.7 ± 0.7 | 92.9 ± 1.3 |
| Spin init. \|+1⟩ fraction (%) | 87.9 ± 1.7 | 70.0 ± 1.9 |
| MW error (%) | 5.6 ± 0.1 | 5.1 ± 0.6 |
| End-to-end fidelity (%) | 88.5 ± 0.5 | 67.1 ± 0.9 |
| Readout fidelity (%) | 96.4 ± 2.2 | 78.6 ± 2.5 |

transitions according to Doherty et al.[10] result in strong transitions separated in the GHz range for high strain upon application of an appropriate magnetic field. In particular, using an aligned magnetic field is very powerful. Also for misaligned magnetic fields (and finite strain), well-separated transitions can be identified (Fig. 4e). In turn, we would estimate 1–2 GHz broad lines as the upper limit. According to Fu et al.[9], this corresponds to around 35 K. Note that we expect the protocol to work with poorly cycling transitions by using a stronger ionization laser.

## Discussion

We have pushed the fidelity of SCC into the single-shot regime, by combining it with resonant excitation at cryogenic temperature. Table 1 summarizes the initialization and readout metrics achieved. The resulting protocol can operate even on shallow implanted NV centers and eliminates the need for any optimized collection optics. We achieve a single-shot SNR of 3.5 and 0.99 on a deep and shallow center respectively, which provides a speedup in the range of $10^3$ over standard readout and a speedup of ~20 over the resonant excitation readout method[7]. As its most important consequence, this technique will enable sensing experiments using long (ms) protocols. These are within the coherence time of shallow NV centers[18], but are currently precluded by acquisition speed; 1 ms of sensing time would enable coherent coupling to a single electron spin at 50 nm distance. In sensing, this would cover the entire thickness of a biological cryoslice[19], in computing it could enable coupling in scalable arrays of NV centers[20]. The protocol is compatible with electric readout of the NV$^-$ spin state[21]; in combination with a single-electron transistor[22], single-shot electric readout might be possible. Our method could also enable single-shot readout of more challenging spin qubits, in particular in silicon carbide, where poor photon count rates currently hamper work for some centers with otherwise promising spin properties[23–25].

## Methods

**Experimental setup.** The measurements were performed in a home-built confocal microscope, with the sample being in a flow cryostat using liquid Helium. Inside the vacuum chamber are a movable permanent magnet and a movable air objective (Nikon Plan Apo 40× NA0.95) to illuminate the sample and to collect fluorescence. The fluorescence was separated from the laser illumination with a 650-nm longpass dichroic mirror. Residual laser light was removed with a 650-nm longpass filter and a shortpass filter (800 nm; just relevant for the ionization with NIR, see SI). Photons were detected with an avalanche photo diode (APD). Three individually gated lasers are combined to a single excitation path, so that the sample can be illuminated simultaneously by all of them: A green 517 nm fiber-pigtailed laser diode driven by a PicoLAS LDP-V 03-100 UF3, 'cleaned-up' with a 540 nm shortpass and combined to the common laser path with a 550 nm longpass dichroic mirror; a red 642 nm fiber-pigtailed laser diode driven by an iC Haus iC-NZN and combined to the external cavity laser's path with a non-polarizing 90:10 beam splitter; a red external cavity diode laser (New Focus TLB-6704) stabilized with a wavemeter (absolute accuracy ±600 MHz) and gated by two acousto-optic modulators (AOM) in series. All laser beams were expanded to ~10 mm, which is approximately the back aperture of the used objective. For each beam, we can control the lateral alignment as well as the collimation. Two field-programmable gate arrays (FPGA) were used to control all short-timescale pulses and to register the APD events.

**Samples preparation.** The deep NV center and the shallow implanted NV center are in two different pieces of diamonds. Both diamonds are electronic grade from Element6 and have some spots where CN$^-$ molecules were implanted. Before implanting, both diamonds were cleaned with a 1:1:1 mixture of sulfuric:nitric:perchloric acid. Afterward, they were annealed at 900 °C for 3 h and cleaned again with the 3-acid mixture. The diamond with the shallow implanted NV was additionally annealed a second time at 1200 °C for 2 h. Before measurements, both diamonds were treated in an Oxygen plasma. To remove high fluorescence in the surrounding of shallow NV centers, we illuminate the region with a high-power (~50 mW) green 532 nm laser after cooling down.

## Data availability

The data that support the findings of this study are available from the corresponding author on reasonable request.

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

## Acknowledgements

We thank Marcus Doherty for helpful discussion. This work has received support from the Deutsche Forschungsgemeinschaft (DFG) under grants RE3606/1-1, RE3606/2-1, RE3606/3-1, and EXC-2111–390814868, from the European Union's Horizon 2020 research and innovation programme under grant agreement No. 820394 (ASTERIQS) and from the National Natural Science Foundation of China (NSFC) under grants 11761131011, 81788101, 91636217, 11722544. D.K. was supported by funds of Polish National Science Center (NCN), PhD Student Scholarship No. 2018/28/T/ST3/00390 and Grant No. 2015/19/B/ST3/03152.

## Author contributions

D.M.I. and F.R. designed the experiment. D.M.I. built the setup. D.M.I., F.P., and F.K. conducted the experiment. D.M.I. and F.P. analyzed the data with support from F.R., F.K., F.S., and J.D. D.M.I. developed all numerical models with support from D.K. and F.P. M.K., T.L., and J.M. prepared the samples. D.M.I., F.P., and F.R. wrote the manuscript. All authors commented on the manuscript.

## Funding

## Competing interests

The authors declare no competing interests.
