## [Peer Review File · Nature Communications]

REVIEWER COMMENTS<

Reviewer #1 (Remarks to the Author):

In this paper, the authors present the single shot readout method based on spin to charge conversion for NV center. Single shot readout is a powerful tool for quantum information technology. For sensing, it greatly boosts up the sensitivity. For quantum computing and communication, it gives the deterministic readouts instead of the probabilistic ones. Before this work, there are two single shot readout protocols in NV center system. Both lead to successful results, but with certain limitations. One utilizes nuclear spin for repetitive readouts, it can work in any temperature and almost with any NV centers. But it requires to use strong magnetic field. The other method uses the spin selective resonant transitions. It only works for the NV centers with good optical spectrum, low strain, and at low temperature. Spin to charge conversion was proposed as an alternative method for the single shot readout. It was believed that this approach can give broader working ranges. This readout has been demonstrated before, but the fidelity was not high enough for reaching the single shot readout level. In this work, the authors cleverly combine the weak resonant laser and a high power below-band laser to realize the high fidelity single shot readout based on spin to charge conversion. This is a huge step in the field. The authors also made systematic analysis and detailed explanations. It is amazing to see that without special optical structure, this single shot readout works well. The paper is well written and deserves the publication in Nature Communications. But still, I have several questions/comments.

1. Usually, by exciting NV centers with the green laser, it can induce spectrum jumps due to its disturbance to the local charge surroundings. This can lead to the shifts of the resonant transitions. Can the authors comment on this effect, since the green laser is used in the beginning of all sequences, as shown in Fig.1 and measurements? After the green pulse, the resonant pulses may become non-resonant. Also, for Fig.2a, have the authors considered this effect to the result? Furthermore, this kind of spectrum jump is quite common in shallow NV centers, even with just resonant red excitation. Does it influence the results?

2. Can the authors comment on the working temperature range of this method? The spin selective transitions will be mixed up when the temperature rises up.

3. Similarly, can the authors also comment on the working range of the strain of NV centers?

4. For the working range of the off-axis magnetic field, could the authors give some estimations?

This is one of the bright spots of this new protocol. Nevertheless, it is worth to analyze this in more details.

For example, the off-axis magnetic field will mix the spin states. This will influence in every part of the protocol, from the initialization to the readout.

Also, in high field, the optical transitions are much more crowded. If the sub-GHz linewidth is taking into account, how large can the magnetic field be?

5. About the Pi pulse for spin initialization step. Since the excited state life time for spin 0 is around 12ns, to reach the spin flips in the ground state, the Pi pulse should be shorter than this time. Was this the case? This fast pulse needs a lot of power.

Also, why not use A1 sub-level resonant transition for the spin initialization?

6. In Fig. 1a, would it be better to mark what MW1, MW2 and 637nm in the level structure? For people working in NV center field, these are obvious. But for people from other fields, seeing labels

in Fig.1c may be confusing.

Reviewer #2 (Remarks to the Author):

The authors have performed a measurement of the NV spin at cryogenic temperatures using a spin to charge conversion sequence based on resonant excitation of one spin sublevel of NV-. The resonant spin-selective excitation enables a highly selective conversion of the spin state to a charge state distribution between NV- and NV0 that can then be measured with high fidelity. This work expands the toolbox of available techniques for NV spin readout, and is of significant interest to the NV community.

The key question of how this method compares to alternative fluorescence based readout protocols remains largely unaddressed in the manuscript. The authors compare their method to readout via detection of fluorescence under off-resonant pumping with green laser light, which pumps all spin states and consequently results in relatively low fluorescence contrast (this is also what limits the contrast of non-resonant spin to charge conversion). However, the state of the art method for spin readout in cryogenic conditions, where resonant excitation of the NV- ZPL is accessible, is rather to selectively drive one spin sublevel (as the authors do here for their resonant spin to charge conversion) while detecting fluorescence. The resonant driving results in a very high contrast (and is also what enables the high contrast for the authors' measurement). So it would be much more meaningful for the authors to directly compare their resonant spin to charge conversion method with resonant fluorescence readout in their setup. This is especially the case given that the resonant fluorescence method is limited by imperfect cycling of the $m_s=0$ spin state, a limit that could potentially be overcome by employing spin to charge conversion, if the ionization step is sufficiently selective. It would greatly expand the impact of this method if it is possible to overcome the cycling limit, but it is not yet clear from the manuscript whether that could be the case. The authors' demonstrated end-to-end fidelity is already quite impressive given the absence of any optical collection enhancements, so a measurement of the resonant fluorescence readout fidelity in their setup would provide a very useful comparison.

A few more minor/technical points:

1. A level of 50kcps is a fair assumption as a "standard" collection efficiency, and seems to be more than adequate for proving the efficacy of the method. I don't see the relevancy of comparing to sub-kcps count rates, as the authors do in their abstract and again in lines 16-21 of p.4.
2. Is the magnetic field particularly chosen to be not aligned to the NV axis? Would it also be possible to apply the protocol with a magnetic field aligned along the NV axis? Why use an off-axis field?
3. There is not much information on spectral diffusion or the degree of power broadening for the deep NV center. Is the NV significantly power broadened for readout and ionization?
4. The charge state initialization sequence for the shallow NV center presented in the supplementary

section S.5 contains first a high-power pulse of 642nm light. In principle, this shouldn't excite any transitions in the NV center. Why is it necessary, or what improvement does it bring over other initialization sequences that were attempted? What sequences were attempted to optimize this initialization?

5. I don't see a specific mention of the 637nm power used for the ionization pulse. Did the authors vary or optimize this power? In any case, it would be good to specify the power used.

6. Lines in some plots are too thick and obscure the relevant details (particularly fig. 1d and insets of figs 2d & 3b). Also the authors could consider plotting the insets to Fig. 2d and 3b on a vertical log scale so that the relevant information is visible.

Brendan Shields

Reviewer #3 (Remarks to the Author):

The ability to read out a quantum state with high fidelity, and ultimately in a single shot is important for implementing quantum protocols. For optically active solid-state qubits, single-shot readout remains challenging due the limited photon flux during the transient spin-dependent fluorescence.

While the NV center in diamond has been extensively studied for many years, the high refractive index of diamond, low photon collection efficiency and limited contrast for spin-dependent fluorescence result in low single-shot SNR for most of the experiments. The conventional spin state readout scheme at room temperatures is based on spin-dependent inter-system crossing, and this scheme requires many repetitions ($\sim 10^5$) to improve the averaged SNR. Previous work has tackled this problem in a number of ways: by using spin-dependent shelving and converting the spin state to charge state, the readout fidelity can be improved. Alternatively, the spin state can also be stored in a nearby nuclear spin and readout repetitively. So far, single-shot readout of NV centers has only been realized using highly cyclic optical transitions at cryogenic temperatures, using a solid immersion lens to improve the optical collection efficiency.

In this work, the authors demonstrate single-shot readout of NV centers by combining two well-established techniques: spin selective resonant excitations and spin-to-charge conversion at cryogenic temperatures. The spin-dependent ionization is achieved by exciting to the excited state with a resonant excitation and ionizing the center from the excited state with a high-power red-detuned laser. Therefore, only the spin-state being addressed by the resonant excitation will be ionized. This work presents the first demonstration of single shot spin readout of NV centers without directly reading out highly cycling optical transitions. This relaxes the conditions for single shot readout significantly.

This work introduces a new protocol for reading out solid-state qubits. While the result is sound and convincing, the presentation of the material is sometimes confusing and difficult to understand, probably especially for people outside the field. I would recommend that the authors address the following points before publication.

A few scientific questions that should be addressed:

1. In the PLE spectra (Fig. 1d and Fig. 4a), there are extra peaks for $|+1\rangle$ spectra that are not present in the simulated spectra. The extra peaks align with peaks in $|0\rangle$ spectra. Is this from imperfect spin initialization with the green laser? If so, I would expect some extra peaks in the $|0\rangle$ spectra that align with peaks in $|+1\rangle$ spectra as well.
2. The authors claim that their use of the resonant laser to read out the charge state is novel. I do not think this claim is correct—see for example, the supplementary information to Andersen et al, Science 364, 154–157 (2019).
3. In Fig. 3a, why is there a decay for the $|+1\rangle$ trace? Is this due to some residual 'off-resonant' excitation of the other transitions with the resonant laser?
4. The authors mentioned that the spectral diffusion is compensated for the shallow NV with higher excitation power (Line 30 - 34 of page 4). However, the postselection on the charge state is accomplished with resonant excitation which in principle should also postselect the transition frequency. The authors should comment on this.
5. Under Section S.3, the authors were not able to saturate the shallow implanted NV with a resonant laser. They attribute this to the spectral diffusion of shallow NV centers. How do they exclude other possibilities such as charge state instability?
6. Why is the spin polarization from green laser quoted differently in the caption of Fig. 2 (85%) and line 18 of page 3 (70%)?
7. In Fig. 4b, the authors attribute the background after 20 us to finite excitation of close by transitions. There is another $|0\rangle$ transition that is further away from other transitions (around -7 GHz shown in Fig. 4a). Why is that transition not used here?

Below are points of clarification and presentation issues to be addressed:

1. The color scheme used in Fig. 1d, Fig. 2a, Fig. 4a and Fig. 4b is confusing. The contrast between $|0\rangle$ and $|+1\rangle$ is not significant enough. Similarly, the inset pulse sequences throughout the paper are too small to be legible.
2. In the inset of Fig. 1d, 'df' is not defined.
3. The inset for Fig. 2a is confusing. The pulse sequence for $|+1\rangle$ initialization should be placed next to the actual data (lower panel). The presentation could also be made clearer by having separate panels for the pulse sequences instead of the indiscriminate uses of insets.
4. The presentation of Fig. 2d and Fig. 3b is confusing. For both figures, the inset for the upper panel has no 'real data' and the y-axis range for the lower panel is cut-off well below the data range. I understand that the authors are trying to show the contrast in the low photon number regime. But it's not informative for the readers to look at an almost blank figure. In this case, it might be better to use log-scale for the y-axis.
5. Line 6 of page 3: what do the authors mean by spin-state stability?
6. Line 12 of page 3: the citation for Robledo et al is missing.
7. In Fig. 4e, the color scheme is confusing, the highlight of $|0\rangle$ transitions is hard to see.
8. Typo: under Section S.1, it should be "... measurements were not able to resolve a more complex hyperfine structure".
9. Typo: the caption of Fig. S1 should be "Hyperfine ODMR transitions ..." instead of "Hyperfine ODRM transitions".

10. Fig. S5 is confusing. The traces (I think) are measured separately while the figure might suggest the traces are measured consecutively.

Reviewer #1 (Remarks to the Author):

In this paper, the authors present the single shot readout method based on spin to charge conversion for NV center. Single shot readout is a powerful tool for quantum information technology. For sensing, it greatly boosts up the sensitivity. For quantum computing and communication, it gives the deterministic readouts instead of the probabilistic ones. Before this work, there are two single shot readout protocols in NV center system. Both lead to successful results, but with certain limitations. One utilizes nuclear spin for repetitive readouts, it can work in any temperature and almost with any NV centers. But it requires to use strong magnetic field. The other method uses the spin selective resonant transitions. It only works for the NV centers with good optical spectrum, low strain, and at low temperature. Spin to charge conversion was proposed as an alternative method for the single shot readout. It was believed that this approach can give broader working ranges. This readout has been demonstrated before, but the fidelity was not high enough for reaching the single shot readout level. In this work, the authors cleverly combine the weak resonant laser and a high power below-band laser to realize the high fidelity single shot readout based on spin to charge conversion. This is a huge step in the field. The authors also made systematic analysis and detailed explanations. It is amazing to see that without special optical structure, this single shot readout works well. The paper is well written and deserves the publication in Nature Communications. But still, I have several questions/comments.

1. Usually, by exciting NV centers with the green laser, it can induce spectrum jumps due to its disturbance to the local charge surroundings. This can lead to the shifts of the resonant transitions. Can the authors comment on this effect, since the green laser is used in the beginning of all sequences, as shown in Fig.1 and measurements? After the green pulse, the resonant pulses may become non-resonant. Also, for Fig.2a, have the authors considered this effect to the result?

Furthermore, this kind of spectrum jump is quite common in shallow NV centers, even with just resonant red excitation. Does it influence the results?

Spectral diffusion is indeed a crucial issue. Our protocol can deal with it as long as line broadening is less than the spacing between different optical transitions, i.e. less than a few GHz. In this condition, the resonant laser will address only one transition, even if its power is chosen sufficiently high to saturate the inhomogeneously broadened line. This constraint is much more relaxed than what is required e.g. for the extraction of indistinguishable photons. In particular, the line can be two orders of magnitude broader than the intrinsic linewidth (~10 MHz). A sub-GHz linewidth for shallow (depth <100 nm) NV centers has been reported several times, once even for centers as shallow as 10 nm (Chu ... Lukin, Nano Letters 14, 1982 (2014)). We extended the discussion of Fig. 4 to address this issue.

Regarding Fig. 2a, spectral diffusion should affect all traces in the same manner, by reducing their intensity, but only their ratio is used to compute initialization fidelity.

2. Can the authors comment on the working temperature range of this method? The spin selective transitions will be mixed up when the temperature rises up.

We expect our method to work up to around 35 K. We added this number at the discussion of Fig. 4e "Simulations of the optical transitions according to Doherty *et al.*¹⁰ result in strong transitions separated in the GHz range. In turn, we would estimate 1-2 GHz broad

lines as the upper limit. According to Fu *et al.*⁹, this corresponds to around 35 K.”

3. Similarly, can the authors also comment on the working range of the strain of NV centers?

We expect the method to work at basically any strain. However, the higher the strain, the more the different transitions tend to get crowded. Nevertheless, upon application of an appropriate magnetic field, a well cycling transition can be separated from other transitions. This is possible for virtually any strain by using an aligned magnetic field. Yet even for misaligned fields, windows of well-isolated and well-cycling transitions will appear, as illustrated by the Figure below - this is a simulation according to Doherty *et al.*¹⁰ for an off-axial strain of 10 GHz and a magnetic field being misaligned by 20° from the NV axis. This figure replaces the former Fig. 4e in the main text. Besides we changed and extended the discussion about the simulation in the main text:

“The protocol also promises to be resilient to high-strain environments, since it does not make use of spin-selective intersystem crossing into the ¹A singlet state. It only requires a well-separated spin-selective transition. Simulations of the optical transitions according to Doherty *et al.*¹⁰ result in strong transitions separated in the GHz range for high strain upon application of an appropriate magnetic field, in particular using an aligned magnetic field is very powerful. Also for misaligned magnetic fields (and finite strain) well-separated transitions can be identified (Fig. 4e). In turn, we would estimate 1-2 GHz broad lines as the upper limit. According to Fu *et al.*⁹ this corresponds to around 35 K. Note that we expect the protocol to work with poorly cycling transitions by using a stronger ionization laser.”

The following Figure replaces the former Fig. 4e. It shows simulation results according to Doherty *et al.*¹⁰ for an off-axial strain of 10 GHz and a magnetic field being misaligned by 20° from the NV axis.

4. For the working range of the off-axis magnetic field, could the authors give some estimations? This is one of the bright spots of this new protocol. Nevertheless, it is worth to analyze this in more details.

For example, the off-axis magnetic field will mix the spin states. This will influence in every part of the protocol, from the initialization to the readout.

Also, in high field, the optical transitions are much more crowded. If the sub-GHz linewidth is taken into account, how large can the magnetic field be?

We expect the method to work with low as well as with high (> 20 GHz) off-axial magnetic field component. However, in between at around 1 to 15 GHz our method will be more difficult to apply because simulated PLE spectra reveal many close-by optical transitions.

One opportunity to improve the working behavior in this intermediate regime is a stronger 642 nm laser for the second photon in the ionization process. A stronger 642nm laser is expected to shorten the ionization time down to just some optical cycles of the NV center. Operating in such a regime would enable the use of the protocol with poorly cycling transitions, e.g. caused by mixing of the spin states. This can also help to overcome the possible limitation of tunneling into the singlet state, which is expected to be more prominent for higher off-axis magnetic field component. The main requirement left would be a well-separated optical transition, which exists for virtually any strain and (off-axial) magnetic field.

Note that the readout might even improve by mixing of spin states as a single resonant laser could act on a A_1 transition simultaneously to pumping on a desired spin zero transition, reducing the need to counteract depletion of the spin state.

We changed Fig. 4e, see answer above, including higher magnetic fields.

5. About the Pi pulse for spin initialization step. Since the excited state life time for spin 0 is around 12ns, to reach the spin flips in the ground state, the Pi pulse should be shorter than this time. Was this the case? This fast pulse needs a lot of power.

Also, why not use A_1 sub-level resonant transition for the spin initialization?

It is important to note that the π -pulse acts on the spin while the NV center is in its optical ground state. To ensure this, there is sufficient time between laser and microwave pulses. Hence, the π -pulse duration can be much slower and is 70ns, as stated in Fig. 3d. To improve clarity of the whole protocol, we added a sketch of the idea below

We agree that the A_1 sub-level resonant transition could have been used for spin initialization, as e.g. demonstrated by Robledo *et al.*⁷ However, due to technical limitations of our setup, we implemented the spin initialization using two MW sources.

6. In Fig. 1a, would it be better to mark what MW1, MW2 and 637nm in the level structure? For people working in NV center field, these are obvious. But for people from other fields, seeing labels in Fig.1c may be confusing.

MW1, MW2 and 637nm are now marked in a slightly revised Fig. 1a.

Reviewer #2 (Remarks to the Author):

The authors have performed a measurement of the NV spin at cryogenic temperatures using a spin to charge conversion sequence based on resonant excitation of one spin sublevel of NV⁻. The resonant spin-selective excitation enables a highly selective conversion of the spin state to a charge state distribution between NV⁻ and NV⁰ that can then be measured with high fidelity. This work expands the toolbox of available techniques for NV spin readout, and is of significant interest to the NV community.

The key question of how this method compares to alternative fluorescence based readout protocols remains largely unaddressed in the manuscript. The authors compare their method to readout via detection of fluorescence under off-resonant pumping with green laser light, which pumps all spin states and consequently results in relatively low fluorescence contrast (this is also what limits the contrast of non-resonant spin to charge conversion). However, the state of the art method for spin readout in cryogenic conditions, where resonant excitation of the NV⁻ ZPL is accessible, is rather to selectively drive one spin sublevel (as the authors do here for their resonant spin to charge conversion) while detecting fluorescence. The resonant driving results in a very high contrast (and is also what enables the high contrast for the authors' measurement). So it would be much more meaningful for the authors to directly compare their resonant spin to charge conversion method with resonant

fluorescence readout in their setup. This is especially the case given that the resonant fluorescence method is limited by imperfect cycling of the $m_s=0$ spin state, a limit that could potentially be overcome by employing spin to charge conversion, if the ionization step is sufficiently selective. It would greatly expand the impact of this method if it is possible to overcome the cycling limit, but it is not yet clear from the manuscript whether that could be the case. The authors' demonstrated end-to-end fidelity is already quite impressive given the absence of any optical collection enhancements, so a measurement of the resonant fluorescence readout fidelity in their setup would provide a very useful comparison.

We appreciate the suggestion to explicitly compare our readout approach to the resonant excitation approach by Robledo *et al.*⁷ Such a comparison has been included in the revised supplementary information.

We therefore made additional histogram measurements for that readout method. For this measurement, we applied the same pulse sequence as used for Fig. 3b except by two modifications to change it to the resonant excitation readout method: We omitted the ionization pulse and removed the simultaneous cw MW excitation during the final readout. Upon varying the readout power and time, we achieved an end-to-end fidelity of 52.8% (SNR: 0.22) for 100 μ s readout duration already including postselection. Clearly in our setup, the resonant excitation readout approach is far from single-shot.

In contrast, our readout scheme (performed directly afterwards in the same setting) results in an end-to-end spin readout fidelity of 83.8% (SNR: 1.38) for a 1ms readout duration. Note that performance is slightly different because strain has changed over time (to \sim 5.7GHz).

In the main text, we added statements in the following sections:

- “Spin-state stability ...”:
“This low number compared to Robledo *et al.*⁷ is due to the fact that we do not use any photonic structures, and precludes direct single-shot readout of the spin by resonant excitation⁷ (fidelity of 52.8%; see Supplementary Fig. S.5 and Discussion S.4).”

- “Spin-Dependent Ionization” (where we also compared our approach to the off-resonant fluorescence approach):
“... and a factor of 20 over resonant excitation readout⁷ for our experimental conditions (see Supplementary Section S.4).”
- “Discussion”:
“... and a speedup of ~20 over the resonant excitation readout method⁷.”

Besides, we extended the “Speed-Up Factor” chapter in the supplementary information by a discussion of the above mentioned experiment, including a histogram showing the count statistics for the resonant excitation readout in our setup.

A few more minor/technical points:

1. A level of 50kcps is a fair assumption as a "standard" collection efficiency, and seems to be more than adequate for proving the efficacy of the method. I don't see the relevancy of comparing to sub-kcps count rates, as the authors do in their abstract and again in lines 16-21 of p.4.

We removed the sub-kcps scenario from the computation of the speedup factor, to avoid exaggerating our claims. However, we sincerely believe that the ability to operate at very low count rates will have a strong impact on the field. The lack of experiments in this regime is rather a symptom of the existing technical limitation than a lack of applications. To mention some specific examples, our method could enable work with single NV centers in an all-fibered setup where a single-mode-fiber is directly bonded to a diamond. It could greatly simplify work on scanning NV centers by enabling side-collection from an AFM tip and by enabling the use of low-NA optics outside the vacuum vessel of a cryostat. It could finally solve an outstanding challenge for color centers in Silicon Carbide, where low fluorescence rates currently are the main remaining roadblock.

2. Is the magnetic field particularly chosen to be not aligned to the NV axis? Would it also be possible to apply the protocol with a magnetic field aligned along the NV axis? Why use an off-axis field?

The field was not aligned due to technical limitations at the setup. We expect that the protocol works well with an aligned magnetic field, too. Simulations according to Doherty *et al.*¹⁰ yield nicely cycling and well isolated optical transitions for aligned magnetic fields, too, see the two plots below (different x-axis scaling) for 1.73 GHz off-axial strain. This holds true even up to high fields (simulated up to 150 GHz ~ 5 T).

We added in Sec. S.1 “We want to stress that this was a non-optimized situation due to technical limitations and a first indication that the field alignment requirements are not strict.” Besides, we added a statement about aligned magnetic fields in the discussion of Fig. 4e: “Simulations of the optical transitions according to Doherty *et al.*¹⁰ result in strong transitions separated in the GHz range for high strain upon application of an appropriate magnetic field, in particular using an aligned magnetic field is very powerful.”

Below is a simulation for a parallel aligned magnetic field and the off-axial strain as measured for the deep NV center (1.73 GHz). The two plots display the same simulation, except for a different scaling on the horizontal axis.

3. There is not much information on spectral diffusion or the degree of power broadening for the deep NV center. Is the NV significantly power broadened for readout and ionization?

The resonant power was optimized on charge-stability during the readout and a high readout fluorescence. According to these criteria, we chose 56 nW, which is indeed below the saturation power ($\sim 10\% P_{\text{sat}}$). However, we performed a control measurement where power broadening just superseded broadening by spectral diffusion. Our protocol also works in this setting, reaching a fidelity of 83.8%. Note that performance is also slightly different because strain has changed over time (to $\sim 5.7\text{GHz}$).

The following plot presents the spin readout count statistics for the control measurement with our protocol, yielding an end-to-end spin readout fidelity of 83.8%. The acquisition time was 1ms and the threshold 2 photons (marked by the dotted blue line).

4. The charge state initialization sequence for the shallow NV center presented in the supplementary section S.5 contains first a high-power pulse of 642nm light. In principle, this shouldn't excite any transitions in the NV center. Why is it necessary, or what improvement does it bring over other initialization sequences that were attempted? What sequences were attempted to optimize this initialization?

We added a sentence in section S.5, hoping to clarify the procedure: “This [the init including red] performed better than a single green initialization pulse with respect to the average fluorescence related to a charge readout afterwards (see above for the

optimization procedure). We attribute this to changing the local surrounding of the shallow implanted NV center. Such changes were demonstrated to shift the charge-state balance.⁹[citation to Dhomkar *et al.*, Nano Letters **18**, 2018]”

5. I don't see a specific mention of the 637nm power used for the ionization pulse. Did the authors vary or optimize this power? In any case, it would be good to specify the power used.

We added a statement about the power in the main text: “The resonant power was kept at 56 nW, as optimized for the readout.”

In fact, we measured Fig. 3a for various resonant powers and saw no pronounced difference when deviating slightly from the optimized readout power.

6. Lines in some plots are too thick and obscure the relevant details (particularly fig. 1d and insets of figs 2d & 3b). Also the authors could consider plotting the insets to Fig. 2d and 3b on a vertical log scale so that the relevant information is visible.

The lines in Fig. 1d are now thinner. The histograms are now plotted on a log y-scale and the insets are omitted.

Reviewer #3 (Remarks to the Author):

The ability to read out a quantum state with high fidelity, and ultimately in a single shot is important for implementing quantum protocols. For optically active solid-state qubits, single-shot readout remains challenging due to the limited photon flux during the transient spin-dependent fluorescence.

While the NV center in diamond has been extensively studied for many years, the high refractive index of diamond, low photon collection efficiency and limited contrast for spin-dependent fluorescence result in low single-shot SNR for most of the experiments. The conventional spin state readout scheme at room temperatures is based on spin-dependent inter-system crossing, and this scheme requires many repetitions ($\sim 10^5$) to improve the averaged SNR. Previous work has tackled this problem in a number of ways: by using spin-dependent shelving and converting the spin state to charge state, the readout fidelity can be improved. Alternatively, the spin state can also be stored in a nearby nuclear spin and readout repetitively. So far, single-shot readout of NV centers has only been realized using highly cyclic optical transitions at cryogenic temperatures, using a solid immersion lens to improve the optical collection efficiency.

In this work, the authors demonstrate single-shot readout of NV centers by combining two well-established techniques: spin selective resonant excitations and spin-to-charge conversion at cryogenic temperatures. The spin-dependent ionization is achieved by exciting to the excited state with a resonant excitation and ionizing the center from the excited state with a high-power red-detuned laser. Therefore, only the spin-state being addressed by the resonant excitation will be ionized. This work presents the first demonstration of single shot spin readout of NV centers without directly reading out highly cycling optical transitions. This relaxes the conditions for single shot readout significantly.

This work introduces a new protocol for reading out solid-state qubits. While the result is sound and convincing, the presentation of the material is sometimes confusing and difficult to understand, probably especially for people outside the field. I would recommend that the authors address the following points before publication.

A few scientific questions that should be addressed:

1. In the PLE spectra (Fig. 1d and Fig. 4a), there are extra peaks for $|+1\rangle$ spectra that are not present in the simulated spectra. The extra peaks align with peaks in $|0\rangle$ spectra. Is this from imperfect spin initialization with the green laser? If so, I would expect some extra peaks in the $|0\rangle$ spectra that align with peaks in $|+1\rangle$ spectra as well.

We added a sentence in the main text before the headline "Spin-state stability ...": "Note that $|0\rangle$ transitions are observed in the $|+1\rangle$ trace because of imperfect spin initialization in combination with the added MW π -pulse."

In the following, we explain this idea in more detail. Imperfect (spin) initialization can be caused by imperfect spin balance and/or imperfect spin reset.

The first issue is a general problem after a green off-resonant pulse for initialization, resulting in roughly 70% spin $|0\rangle$ and 15% in $|+1\rangle$ in our case. The MW π -pulse after the green init pulse is meant to flip $|0\rangle$ to $|+1\rangle$. However, it works in both directions simultaneously, i.e. the π -pulse inverts the populations - afterwards there is a $\sim 15\%$ fraction still/again in $|0\rangle$. This argument is just partially accountable for the questioned effect, because otherwise the $|+1\rangle$ peaks should be visible in the $|0\rangle$ spectrum, too, as noted by the referee.

The second issue is imperfect spin reset, which is related to both questions of the referee. Imperfect spin reset is relevant because the resonant laser mixes/depletes the currently addressed spin state and the green laser pulse is shorter than for the other data presented (500ns instead of 2 μ s) and weaker (0.4mW instead of 1.4mW). Hence, we expect the reset to not take place in each repetition of the laser pulse sequence. The “soft” green init is chosen on purpose to yield a “cleaner” PLE spectrum.

When the resonant laser is tuned to a spin $|0\rangle$ transition, it depletes the $|0\rangle$ population and shifts it to $|\pm 1\rangle$. In turn the fluorescence vanishes and spin-reset is necessary, which is performed by the green laser pulse. As this pulse is “soft”, it may not reset the spin state in each repetition. In the trace without an MW π -pulse ($|0\rangle$ trace; “right” trace for spin $|0\rangle$), this simply causes some reduction in measured fluorescence. In the trace with π -pulse ($|+1\rangle$; “wrong” trace for spin $|0\rangle$), the π -pulse can act instead of the green laser pulse: After shifting the spin population to $|\pm 1\rangle$ and a failed green reset, the π -pulse will switch the $|+1\rangle$ population to $|0\rangle$ and in turn restore some fluorescence.

In case of the resonant laser being tuned to a $|+1\rangle$ transition, this second effect is not available in the “wrong” trace as there is no π -pulse included. Hence, this transition is depleted quickly, while the green laser does not restore the fluorescence to an extent that the fluorescence by this transition is above noise level. Note that for $|\pm 1\rangle$ transitions, the spin mixing under resonant optical pumping is stronger because cycling within the optical transition is expected to be less.

2. The authors claim that their use of the resonant laser to read out the charge state is novel. I do not think this claim is correct—see for example, the supplementary information to Andersen *et al.*, *Science* 364, 154–157 (2019).

We now cite the SI for Andersen *et al.*¹² and changed the second part of the phrase: “We also use the resonant laser to read out the charge state¹²[citation to SI for Andersen *et al.*], which is in contrast to *most* SCC publications so far, ...”.

We see that Andersen *et al.*¹² used a resonant 637 nm laser for charge-state detection. However, they do not mention any means to counteract spin depletion, which should require low resonant power. In contrast, we counteract spin mixing by simultaneous cw MW, which enables us to collect two orders of magnitude more photons compared to resonant pumping alone. This is, in particular, the key factor to efficiently read out the charge state for low count rates as present in our case.

3. In Fig. 3a, why is there a decay for the $|+1\rangle$ trace? Is this due to some residual ‘off-resonant’ excitation of the other transitions with the resonant laser?

We added a sentence in the main text: “We expect the decay for spin $|+1\rangle$ for long ionization times to stem from residual excitation by the 642 nm laser.”

In general, we do see three possible reasons for this decay: 1) Excitation of a close-by transition with different spin character, 2) residual 642nm laser light at lower wavelengths promoting the first photon, too, and 3) effects to the surrounding by the 642nm laser. To prove the hypothesis of excitation by the 642nm laser, we performed two additional measurements (not shown in main text or supplementary information).

- a) We performed the measurement type presented in Fig. 3a without resonant illumination. We observed this decay at long ionization times, too.
- b) We took a spectrum of the 642 nm diode laser emission. We observed lower-wavelength components in the emission.

The red line in the plot below indicates 637.2nm

4. The authors mentioned that the spectral diffusion is compensated for the shallow NV with higher excitation power (Line 30 - 34 of page 4). However, the postselection on the charge state is accomplished with resonant excitation which in principle should also postselect the transition frequency. The authors should comment on this.

Indeed, we cannot exclude that postselection also removes events of severe spectral diffusion. We now explicitly state this in the revised manuscript in the “Charge-state” section: “Postselection also removes repetitions with severe spectral diffusion.”

Still, we emphasize that the fraction of accepted events in our postselection is actually much higher than for previous demonstrations of single-shot-readout (37% for us as compared to 2-5% for Robledo *et al.*⁷).

For a sufficiently narrow line, the protocol can work in a regime where the resonant line is fully saturated so that postselection only targets fluctuations of the charge state. We performed a control experiment on the deep NV center to verify this claim, reaching a fidelity of 83.8% in the fully saturated regime (postselecting on 49% of repetitions). Note that performance is also slightly different because strain has changed over time (to ~5.7 GHz).

The revised manuscript is also more explicit on how the parameters have been optimized. “Spectral lines are inhomogeneously broadened to (0.43 ± 0.02) GHz due to spectral diffusion (see Supplementary Section S.10). As the transitions are also much weaker - yielding lower fluorescence - we compensate for this by increasing the resonant red laser power to 240 nW, which is just low enough to prevent pronounced effects by also exciting the close-by $|1\rangle$ transition. Still, this close-by transition compromises spin initialization fidelity (Fig. 4b).”

Note that for both the deep and the shallow NV center presented we did not choose the power according to saturation of the used transition. Instead, we optimized the power to yield as high (average) fluorescence as possible in a charge-state readout while, firstly, just exhibiting a negligible or no decay in fluorescence over readout time and, secondly, not exciting close-by $|1\rangle$ transitions.

5. Under Section S.3, the authors were not able to saturate the shallow implanted NV with a resonant laser. They attribute this to the spectral diffusion of shallow NV centers. How do they exclude other possibilities such as charge state instability?

In case of charge state instability, we expect that increasing the 637 nm laser power will shift the charge state balance towards NV^0 because 637 nm can nicely ionize NV^- to NV^0 , but cannot cause the other way round due to the NV^0 zero-phonon transition being larger in energy (575 nm). In turn, we exclude charge state instability as a source for this non-saturation behavior.

We added some sentences at the end of section S.3 discussing this issue:

“Note that in case of charge instability, increasing the resonant power is expected to shift the charge state balance more and more towards the dark NV^0 charge state as 637nm illumination can cause ionization to NV^0 but cannot cause recombination from NV^0 . Hence, we exclude this effect as a source for the non-saturation behavior; in particular it is even a slight indication that the charge state seems to be reasonably stable in our case.”

6. Why is the spin polarization from green laser quoted differently in the caption of Fig. 2 (85%) and line 18 of page 3 (70%)?

The numbers in the text represented the fraction of events for the respective spin state. The numbers in the caption was the spin initialization fidelity. We now state the fraction at all occurrences in the main manuscript.

7. In Fig. 4b, the authors attribute the background after 20 us to finite excitation of close by transitions. There is another $|0\rangle$ transition that is further away from other transitions (around -7 GHz shown in Fig. 4a). Why is that transition not used here?

We tried the same procedure with that line, and got similar results. The end-to-end fidelity was a bit higher and the corrected fidelity a bit lower.

Below are points of clarification and presentation issues to be addressed:

1. The color scheme used in Fig. 1d, Fig. 2a, Fig. 4a and Fig. 4b is confusing. The contrast between $|0\rangle$ and $|+1\rangle$ is not significant enough. Similarly, the inset pulse sequences throughout the paper are too small to be legible.

In Fig. 1d, 2a, 4a and 4b light blue was changed to magenta. The insets were enlarged as far as possible within the axes.

2. In the inset of Fig. 1d, 'df' is not defined.

“df indicates a change of the laser detuning.” was added in the figure caption.

3. The inset for Fig. 2a is confusing. The pulse sequence for $|+1\rangle$ initialization should be placed next to the actual data (lower panel). The presentation could also be made clearer by having separate panels for the pulse sequences instead of the indiscriminate uses of insets.

For Fig. 2a, the insets are now within the corresponding axis systems.

As we were often missing information about pulse sequences in other publications, we wanted to include as much as possible for the interested reader without taking up too much space.

4. The presentation of Fig. 2d and Fig. 3b is confusing. For both figures, the inset for the upper panel has no 'real data' and the y-axis range for the lower panel is cut-off well below the data range. I understand that the authors are trying to show the contrast in the low photon number regime. But it's not informative for the readers to look at an almost blank figure. In this case, it might be better to use log-scale for the y-axis.

Fig. 2d, 3b and 4d are now without the insets and the main axes have logarithmic y axis scaling.

5. Line 6 of page 3: what do the authors mean by spin-state stability?

The paragraph title was changed to “Spin-state stability under optical pumping and spin-state initialization”.

6. Line 12 of page 3: the citation for Robledo et al is missing.

We added the citation.

7. In Fig. 4e, the color scheme is confusing, the highlight of $|0\rangle$ transitions is hard to see.

The highlighting is now more intense.

8. Typo: under Section S.1, it should be "... measurements were not able to resolve a more complex hyperfine structure".

We corrected the typo.

9. Typo: the caption of Fig. S1 should be "Hyperfine ODMR transitions ..." instead of "Hyperfine ODRM transitions".

We corrected the typo.

10. Fig. S5 is confusing. The traces (I think) are measured separately while the figure might suggest the traces are measured consecutively.

In fact, the traces are measured interleaved, i.e. all "partial traces" (as indicated by coloring in Fig. S.5) are measured consecutively before the full sequence (comprising all 12 partial sequences) repeats for the stated 500 000 (200 000) repetitions. This leads to possible long-term effects affecting all partial sequences similarly, so that they are comparable to each other.

We added a statement in the caption of Fig. S.5; and also for Fig. S.4 as it displays the same data.

REVIEWERS' COMMENTS

Reviewer #1 (Remarks to the Author):

As stated in my previous report, this work is landmark one in the field. The authors have sincerely responded to my questions. I recommend the publication of the paper in Nature Communications.

Reviewer #2 (Remarks to the Author):

The authors have addressed all of my concerns and I find the work to be compelling and of wide interest.

A couple of minor points remained in my reading of the manuscript:

1. The SNRs quoted in the introduction seem to be for 200kcps and, I think, 50kcps, rather than 5kcps?
2. A very relevant and complementary article [1] has appeared on the arXiv in the time since this manuscript was originally submitted. I think it would be worth pointing readers to that work as well.

[1] Q. Zhang et al., High-fidelity single-shot readout of single electron spin in diamond with spin-to-charge conversion, arXiv:2009.14172 [quant-ph]

Reviewer #3 (Remarks to the Author):

The authors have addressed all of my concerns and comments, and I think the manuscript is suitable for publication.

Dear Editor,

We are delighted to hear that our manuscript got accepted to be published in Nature Communications.

Once more, we want to thank the reviewers for their valuable comments, which lead to many improvements of our manuscript.

We addressed the remaining two small issues as described below. Track changes was activated for the manuscript.

Reviewer #1 (Remarks to the Author):

As stated in my previous report, this work is landmark one in the field. The authors have sincerely responded to my questions. I recommend the publication of the paper in Nature Communications.

We thank the reviewer for the positive comment.

Reviewer #2 (Remarks to the Author):

The authors have addressed all of my concerns and I find the work to be compelling and of wide interest.

We thank the reviewer for the positive comment.

A couple of minor points remained in my reading of the manuscript:

1. The SNRs quoted in the introduction seem to be for 200kcps and, I think, 50kcps, rather than 5kcps?

The reviewer is right. We corrected for the missing 0; it is now 200 kcps and 50 kcps.

2. A very relevant and complementary article [1] has appeared on the arXiv in the time since this manuscript was originally submitted. I think it would be worth pointing readers to that work as well. [1] Q. Zhang et al., High-fidelity single-shot readout of single electron spin in diamond with spin-to-charge conversion, arXiv:2009.14172 [quant-ph]

We included the citation to this preprint at two occasions in the paper: When discussing the charge-state readout using the resonant laser and when discussing ionization with NIR.

Reviewer #3 (Remarks to the Author):

The authors have addressed all of my concerns and comments, and I think the manuscript is suitable for publication.

We thank the reviewer for the positive comment.